# Conventional and Unconventional Therapeutic Strategies for Sialidosis Type I

**DOI:** 10.3390/jcm9030695

**Published:** 2020-03-04

**Authors:** Rosario Mosca, Diantha van de Vlekkert, Yvan Campos, Leigh E. Fremuth, Jaclyn Cadaoas, Vish Koppaka, Emil Kakkis, Cynthia Tifft, Camilo Toro, Simona Allievi, Cinzia Gellera, Laura Canafoglia, Gepke Visser, Ida Annunziata, Alessandra d’Azzo

**Affiliations:** 1Department of Genetics, St. Jude Children’s Research Hospital, Memphis, TN 38105, USA; rosario.mosca@stjude.org (R.M.); diantha.vandevlekkert@stjude.org (D.v.d.V.); yvan.campos@stjude.org (Y.C.); lfremuth@stjude.org (L.E.F.); ida.annunziata@stjude.org (I.A.); 2Department of Anatomy and Neurobiology, College of Graduate Health Sciences, University of Tennessee Health Science Center, Memphis, TN 38163, USA; 3Ultragenyx Pharmaceutical, Novato, CA 94949, USA; JCadaoas@ultragenyx.com (J.C.); vkoppaka@ultragenyx.com (V.K.); EKakkis@ultragenyx.com (E.K.); 4Office of the Clinical Director & Medical Genetics Branch, National Human Genome Research Institute, National Institutes of Health (NHGRI), Bethesda, MD 20892, USA; cynthiat@mail.nih.gov; 5Undiagnosed Disease Network, National Human Genome Research Institute, National Institutes of Health, Bethesda, MD 20892, USA; toroc@mail.nih.gov; 6Unit of Genetics of Neurodegenerative and Metabolic Diseases, Fondazione IRCCS Istituto Neurologico Carlo Besta, 20133 Milan, Italy; simona.allievi@istituto-besta.it (S.A.); cinzia.gellera@istituto-besta.it (C.G.); 7Neurophysiopathology, Fondazione IRCCS Istituto Neurologico Carlo Besta, 20133 Milan, Italy; laura.canafoglia@istituto-besta.it; 8Department of Metabolic Diseases, Wilhelmina Children’s Hospital, University Medical Center Utrecht, 3584 CX Utrecht, The Netherlands; G.Visser-4@umcutrecht.nl

**Keywords:** sialidosis type I, NEU1, PPCA, dietary and pharmacological compounds, therapy

## Abstract

Congenital deficiency of the lysosomal sialidase neuraminidase 1 (NEU1) causes the lysosomal storage disease, sialidosis, characterized by impaired processing/degradation of sialo-glycoproteins and sialo-oligosaccharides, and accumulation of sialylated metabolites in tissues and body fluids. Sialidosis is considered an ultra-rare clinical condition and falls into the category of the so-called orphan diseases, for which no therapy is currently available. In this study we aimed to identify potential therapeutic modalities, targeting primarily patients affected by type I sialidosis, the attenuated form of the disease. We tested the beneficial effects of a recombinant protective protein/cathepsin A (PPCA), the natural chaperone of NEU1, as well as pharmacological and dietary compounds on the residual activity of mutant NEU1 in a cohort of patients’ primary fibroblasts. We observed a small, but consistent increase in NEU1 activity, following administration of all therapeutic agents in most of the fibroblasts tested. Interestingly, dietary supplementation of betaine, a natural amino acid derivative, in mouse models with residual NEU1 activity mimicking type I sialidosis, increased the levels of mutant NEU1 and resolved the oligosacchariduria. Overall these findings suggest that carefully balanced, unconventional dietary compounds in combination with conventional therapeutic approaches may prove to be beneficial for the treatment of sialidosis type I.

## 1. Introduction

Neuraminidase 1 (NEU1) is a lysosomal sialidase that initiates the hydrolysis of glycoproteins and oligo- or polysaccharides by removing terminal sialic acid residues from the non-reducing end of their glycan chains. NEU1 forms a high molecular weight complex with two other lysosomal enzymes, the acidic β-galactosidase (β-GAL) and the serine carboxypeptidase, protective protein/cathepsin A (PPCA) [1]. In contrast to β-GAL that partially retains its activity outside the complex, NEU1 strictly depends on its association with PPCA for proper folding, catalytic activation and stability in lysosomes [1,2]. In absence of a functional PPCA, NEU1 activity is no longer measurable and the enzyme is rapidly degraded. The importance of NEU1’s function for proper lysosomal catabolism and maintenance of cell and tissue homeostasis is exemplified by the severe systemic and neurological consequences of NEU1 deficiency in patients with the lysosomal storage diseases (LSDs) sialidosis, primary deficiency of NEU1, and galactosialidosis secondary deficiency of NEU1 due to a primary defect in PPCA [3,4]. Loss of NEU1 enzymatic activity affects the lysosomal processing/degradation of several sialo-glycoconjugates, and results in the gradual accumulation of sialylated metabolites in tissues and in urinary excretion of sialyloligosaccharides, diagnostic hallmarks of both diseases.

Based on the age of onset and severity of the clinical symptoms, sialidosis is loosely classified into two subtypes: type II the early onset, dysmorphic form, and type I, the normomorphic, attenuated form [3]. Type II sialidosis occurs in patients either at birth with a congenital, fulminant disease associated with hydrops faetalis, ascites and early death, or within the first year of life with symptoms including coarse face, enlargement of spleen and liver, dysostosis multiplex, vertebral deformities, and severe mental retardation. Type I sialidosis, also referred to as cherry red spot, myoclonus syndrome, has quite a different clinical course. Patients are mostly asymptomatic until late childhood, when they begin to show signs of myoclonus, seizures, ataxia and visual impairment [5]. These clinical manifestations range broadly and do not always correlate with the *NEU1* mutations involved. Type I patients have a longer life expectancy and normal intellectual abilities, but can develop severe myoclonus, ataxia, inability to deambulate and speech impairment, and may become wheelchair-bound as the disease progresses [5,6,7,8,9]. Mild symptoms, especially at the time of the first occurrence, are often indistinguishable from those associated with other neurosomatic conditions, resulting in patients being misdiagnosed or diagnosed years after their first clinical problems [5]. Consequently, families of sialidosis type I have frequently more than one sibling affected [8]. With the advent of whole genome or exome sequencing, several novel pathogenic NEU1 mutations have been identified in homozygosity or compound heterozygosity, even in patients with no clinical or biochemical features characteristic of sialidosis, like oligosacchariduria, which poses an additional complication for diagnosis [8]. However, molecular analysis has indeed enabled the early diagnosis of new cases and their number increases every year. Thus, it is becoming more and more clear that the incidence of sialidosis type I in the general population is higher than anticipated for an “orphan” disease (1:250,000 to 1:2,000,000 live births [5]) for which only palliative care is currently available, but unfortunately no target therapy.

Animal models of both sialidosis type I and II have been generated [10,11]. These mouse models are faithful to the sialidosis types they represent; *Neu1^−/−^* mice with symptoms at birth develop a severe, systemic disease, affecting most visceral organs, the heart, muscle and the nervous system, and is associated with progressive edema and oligosacchariduria [10]. In contrast, the *NEU1^−/−^; Neu1^V54M^* mice, carrying a single amino acid substitution (V54M) found in patients with type I sialidosis, mimic the type I form of the disease; they are viable and fertile with normal gross appearance and develop mild histopathology, particularly in the kidney, and oligosacchariduria between 1–2 years of age [11]. Canonical therapeutic approaches, including enzyme replacement therapy (ERT) [12], pharmacologic chaperone therapy with PPCA, and self-complementary adeno-associated virus (scAAV)-mediated gene therapy [11] have been tested successfully in both *Neu1^−/−^* and *NEU1^−/−^; Neu1^V54M^*. A short-term ERT was performed in *Neu1^−/−^* mice using a recombinant Neu1 enzyme purified from overexpressing insect cells [12]. Although this treatment led to increased Neu1 enzymatic activity and widespread correction of the pathological signs in many of the visceral organs, the recombinant enzyme was highly immunogenic in the knockout mice and elicited a severe immunological response that hampered long-term assessments of this therapeutic approach [12]. In contrast, a chaperone-mediated therapy was tested successfully in *NEU1^−/−^; Neu1^V54M^*. Systemic injection of these mice with a single dose of an adeno-associated viral vector expressing PPCA [13], the natural chaperone of NEU1, resulted in high expression of PPCA in the liver of the injected mice, which, in turn, enhanced Neu1 activity in all tested tissues and reverted kidney pathology and oligosacchariduria [11,13]. Although these therapeutic approaches hold great promise for the treatment of sialidosis patients, none of them are currently available in the clinic. Given the growing number of patients with type I sialidosis, and the urgent need for treatments, it is crucial to look outside the box and experiment not only with conventional therapies but also with alternative ways to halt or ameliorate disease symptomatology. In line with this notion is the recent finding that the histone deacetylase (HDAC) inhibitor SAHA enhances *NEU1* mRNA expression and NEU1 residual activity in fibroblasts from patients with both type I and type II sialidosis [14]. It is likely that other genetic/epigenetic modifiers as well as environmental factors, including specific diet regimens, and food supplements, could influence the levels of residual enzyme activity and the penetrance of specific phenotypes. This reasoning is supported by the successful results obtained by administering the dietary supplement, betaine (trimethylglycine), to fibroblasts of patients with aspartylglucosaminuria (AGA), another orphan lysosomal disorder prevalent in Finland but rare worldwide and for which there is no therapy [15]. These authors have shown that betaine can act as a pharmacological chaperone, increasing the residual activity of mutant aspartylglucosaminidase when administered to fibroblasts from patients with AGA. Although the actual mechanism of action of betaine in a lysosomal disease is not clearly understood, these findings have spearheaded a clinical trial with this compound in Finnish patients.

In this study we wanted to test and compare the effects of conventional and unconventional therapeutics, i.e., a recombinant form of human PPCA (rhPPCA), and pharmacological and dietary compounds, on the residual activity of NEU1 mutant proteins in a cohort of 12 primary fibroblast strains isolated from type I sialidosis patients and in mice with residual Neu1 activity, mimicking sialidosis type I. This is in an effort to identify alternative therapeutic approaches that could improve the quality of life of patients with this condition.

## 2. Materials and Methods

### 2.1. Human Fibroblasts

Skin fibroblasts from control individuals were obtained from Coriell Institutes. Human fibroblasts from patients with type I sialidosis and from a patient with galactosialidosis were uncoded and unidentifiable and were obtained from the Pediatric Undiagnosed Diseases Program, National Human Genome Research Institute/NIH (Bethesda, MD, USA), from the I.R.C.C.S., Istituto Neurologico Carlo Besta, Milan, Italy and from the Univesitair Medische Centrum Utrecht, Utrecht, The Netherlands. Original consent was obtained by the clinicians from the patient or a family member and the study was approved by the ethics committees of the three institutions. Cells were banked for secondary future research. All fibroblasts were cultured and maintained in DMEM-FBS complete medium (10% fetal bovine serum (Gibco, Gaithersburg, MD, USA), 2 mM Glutamax (Gibco), penicillin (100 U/mL), and streptomycin (100 mg/mL) (Gibco) at 37 °C/5% CO_2_.

### 2.2. Production and Purification of Human Recombinant PPCA Protein

rhPPCA was produced by Ultragenyx Pharmaceutical. cDNA encoding human protective protein/cathepsin A (PPCA; 54 kDa precursor) subcloned into the mammalian expression vector pFN10A(ACT) Flexi^®^ vector (Promega, Madison, WI, USA), was transfected into chinese hamster ovary (CHO) cells. CHO cells were grown in suspension in a chemically defined, protein-free medium Acti CHO medium (GE Healthcare Life Sciences, Pittsburgh, PA, USA) and the highest expression clone was expanded and adapted to suspension cultures in production medium (GE Healthcare Life Sciences). rhPPCA protein was purified on a chromatography column, followed by diafiltration.

### 2.3. Treatment of Fibroblasts with rhPPCA, Romidepsin or Betaine

For PPCA treatment, fibroblasts were seeded at a density of 2 × 10^5^ cells per well in a 6-well plate in triplicate. Cells were cultured for 1 day and treated for 48 h with human PPCA at a concentration of 1 μg/mL in DMEM-complete medium.

For romidepsin or betaine treatment, fibroblasts were seeded into 10 cm dishes at a density of 5 × 10^5^ cells per dish. The cells were treated with 100 mM betaine (Sigma Aldrich, St. Louis, MO, USA) for 48 h or with 10 nM romidepsin (Active Motif, Carlsbad, CA, USA) for 24 h.

### 2.4. RNA Isolation and Real-Time Quantitative PCR

Total RNA was isolated from primary human skin fibroblasts from healthy individuals and sialidosis fibroblasts using the PureLink RNA Kit (Life Technologies, Carlsbad, CA, USA) according to the manufacturer’s protocol. DNA contaminants were removed on a DNAse I column (Life Technologies), according to the manufacturer’s protocol. RNA quantity and purity were measured using NanoDrop Lite spectrophotometer (Thermo Fisher Scientific, Carlsbad, CA, USA). Complementary DNA was produced using 3 μg of total RNA with RT2 First Strand Kit (Qiagen, Germantown, MD, USA). RT-qPCR was performed using RT2 SYBR Green qPCR Mastermix, 1 μL (100 ng) of cDNA, 10 μM primer (*NEU1* qHsaCED0037311 and *HPRT1* qHsaCID0016375; Bio-Rad, Hercules, CA, USA), and RNAse-free water in a 25 μL reaction volume on a CFX96 real-time PCR machine (Bio-Rad). Samples were normalized to *HPRT1*. The plotted values represent the relative normalized expression of the mRNA in betaine- or romidepsin-treated cells compared to untreated cells.

### 2.5. Western Blot Analyses

Cells were lysed with a hypotonic shock. Lysates (10 μg) underwent electrophoresis on Criterion TGX precast gels (10 or 12%; Bio-Rad), gels were blotted against PVDF membranes and blocked in 5% nonfat dry milk. rhPPCA protein (2 and 6 ug) was loaded on Criterion TGX precast gels (10% Bio-Rad), gels were blotted against Polyvinylidene Difluoride (PVDF) membranes. The latter was stained with Coomassie brilliant Blue (Bio-Rad). Membranes were immunoblotted for anti-NEU1 antibody (in house prepared 1:300) and anti-PPCA antibody (in house prepared 1:250), anti-H3 antibody (Cell Signaling #4499 1:1000) and anti-H3K4me3 antibody (Abcam ab8580 1:500) and incubated overnight at 4 °C in 3% BSA–TBS-T solution. The next day, membranes were incubated with HRP-conjugated secondary antibodies and developed using SuperSignal West Femto Maximum Sensitivity Substrate or SuperSignal West Pico (Thermo Fisher Scientific, Carlsbad, CA, USA). Quantitative analyses of the immunoblots were performed with Image Lab software.

### 2.6. Animals and Betaine Treatment

Animals were housed in a fully AAALAC (Assessment and Accreditation of Laboratory Animal Care) accredited animal facility with controlled temperature (22 °C), humidity, and lighting (alternating 12-h light/dark cycles). Food and water were provided *ad libitum*. All procedures in mice were performed according to animal protocols approved by the St. Jude Children’s Research Hospital Institutional Animal Care and Use Committee and National Institutes of Health guidelines. *Neu1^−/−^; NEU1^V54M^* were described in Bonten et al. 2013 [11]. *NEU1^del654-659^* mice were generated via CRISPR/Cas9 technology, which led to an in-frame deletion. Briefly, the sgRNA CGG GAC GCT GGA GCG AGA was microinjected in oocytes from mice expressing Cas9 (Jackson Laboratories, Bar Harbor, ME). The offspring was sequenced for mutations and the alleles were segregated. Betaine (Sigma Aldrich) was supplemented in the drinking water at a concentration of 1.5% (w/v). *Neu^+/−^; NEU1^V54M^* (1 month) *Neu1^−/−^; NEU1^V54M^* (1 months), *WT* (1 year) and *NEU1^del654-659^* (1 year) animals were treated with betaine for 1 month.

### 2.7. Peripheral Blood Mononuclear Cells (PBMC) Isolation

Mice were anesthetized with Avertin (12.5 mg/mL) at 20 μL/g body weight. Whole blood was collected in tubes containing 10 μL 250 mM Ethylenediaminetetraacetic acid (EDTA) (in house solution from St. Jude Children’s Research Hospital (STJCRH) veterinarian pathology core). 500 μL of whole blood was mixed with 20 mL DPBS/2 mM EDTA and layered on top of 5 mL 70% standard isotonic Percoll solution (Sigma Aldrich). The blood was centrifuged in a swing bucket rotor at 400× *g* at 20 °C for 40 min. The peripheral blood mononuclear cells (PBMC) layer was carefully removed from the tubes and transferred to new 50 mL tubes. The PBMCs were washed in 35 mL with DPBS/2 mM EDTA buffer and centrifuged at 500× *g* at 20 °C for 10 min. The cell pellets were re-suspended in 1 mL DPBS/2 mM EDTA buffer and transferred into 1.7 mL Eppendorf tubes. The PBMCs were recovered with a final centrifugation at 5000× *g* at 20 °C for 2 min and processed immediately for enzyme activities.

### 2.8. NEU1 and Cathepsin A Activity Assays

NEU1 and cathepsin A catalytic activities were measured against synthetic substrates, 2′-(4-methylumbelliferyl)-α-D-N-acetylneuraminic acid, sodium salt, and the dipeptide Z-Phe-Ala. Briefly, human sialidosis fibroblasts, mouse PBMCs and kidneys from betaine treated mice were lysed in ddH_2_O containing 0.5% triton-X-100. For NEU1 enzymatic activity, 5 μL cell/tissue homogenate was incubated with 5 μL substrate in triplicate in 96-well plates at 37 °C for 1 h. The reaction was stopped by addition of 200 μL 0.5 M carbonate buffer, pH 10.7. For carboxypeptidase activity of cathepsin A 10 μL of homogenates were incubated with 90 μL of substrate solution consisting of 1.5 mM *N*-carbobenzoxy–l–phenylalanyl–l–alanine (Z-Phe-Ala) in 50 mM 4-morpholineethanesulfonic acid buffer (MES; pH 5.5). The sealed plates were incubated at 37 °C for 30 min followed by boiling on a 100 °C heating block for 5 min. Alanine reagent was freshly prepared by adding 500 μL of *o*-phthaldialdehyde (10 mg/mL in 96% ethanol) and 500 μL of β-mercaptoethanol (5 μL/mL in 96% ethanol) to 30 mL of 50 mM borate buffer, pH 9.5. Ten L per sample was transferred to a new 96-well plate and incubated with 300 μL of alanine reagent for 10 min. The fluorescence was measured (EX-355, EM-460) and the specific enzyme activities were calculated. NEU1 activities were calculated as nanomoles of substrate converted per hour per milligram of protein (nmol/mg/h), whereas the cathepsin A activities were calculated as picomoles of substrate converted per minute per milligram of protein (pmol/mg/min).

### 2.9. Sialic Acid Assay

The total sialic acid content in mouse urine was measured using the Enzychrom Sialic Acid Assay Kit, according to the manufacturer’s instructions (BioAssay Systems, Hayward, CA, USA).

### 2.10. Statistical Analyses

Statistical analyses were performed using GraphPad Prism. Quantitative data are presented as mean ± SD. For comparisons between two groups, Student’s *t*-test (unpaired, two-tailed) was performed. Groups were considered different when *p* < 0.05. For all quantifications, a minimum of three independent experiments were performed. Measurements were taken from distinct samples. The number of replicates is noted in the figure legends.

## 3. Results

### 3.1. Characterization of Fibroblasts from Type I Sialidosis Patients

The primary fibroblasts used in this study were derived from 12 patients with confirmed diagnosis of sialidosis type I. A list of the *NEU1* mutations identified in these patients and their position in the primary structure of the protein are given in Figure 1A,B. For six of the patients, the mutations have been reported earlier [8,16,17,18,19]. The remaining six patients, not described in the literature, carry novel *NEU1* mutations (p.Ala167Val, p.Tyr268Cys, p.Ser410Arg-fs), as well as a splice variant G > A at intron 3 + (g.1635G > A-fs), five of which in compound heterozygosity with mutations shared by other sialidosis patients (Figure 1B).

All patients’ fibroblasts expressed different levels of *NEU1* mRNA, as determined by quantitative real-time PCR analysis, some even higher than wild-type (WT) levels (Figure 1C). Normal or high levels of *NEU1* mRNA did not, however, correlate with NEU1 residual enzyme activity measured in the patients’ fibroblasts that ranged between ~0.6–4% of controls (Figure 1D). Immunoblot analysis of total fibroblast lysates probed with affinity purified anti-NEU1 antibody revealed the presence of NEU1 protein variants of ~46 kDa in all the patients’ samples, albeit the amounts were markedly reduced in most of the patients’ lysates compared to normal fibroblasts (Figure 1E and Appendix A).

### 3.2. Mutant NEU1 Enzyme Activity is Enhanced by Exogenous rhPPCA in Type I Sialidosis Fibroblasts

It was shown previously that the amounts of PPCA protein available to complex with NEU1 could be rate limiting, and that the basal activity of WT NEU1 could be increased by exogenously incrementing PPCA levels [16]. It was also demonstrated that NEU1 activity in *NEU1^−/−^; Neu1^V54M^* mice responds to PPCA levels in a concentration-dependent manner [11,16]. We now wanted to assess the generality of this approach by performing an in vitro ERT in this large cohort of fibroblasts from type I sialidosis patients. Recombinant human PPCA (rhPPCA) was produced in Chinese hamster ovary (CHO) cells overexpressing the precursor-form of human PPCA and was purified to homogeneity from the conditioned medium (Appendix A). To test whether the rhPPCA was efficiently internalized and delivered to the lysosomes, we first treated human fibroblasts isolated from one patient with galactosialidosis. As expected, uptake of rhPPCA restored cathepsin A activity to control levels and led to consequent normalization of NEU1 activity (Figure 2A,B). Next, we performed similar uptake experiments in all type I sialidosis fibroblasts. As shown in Figure 2C, treatment of sialidosis fibroblasts with rhPPCA resulted in a modest but consistent increase of the NEU1 residual activity in several of the patients’ cells. However, these levels were significantly different only for four of the patients’ fibroblasts (Figure 2C). Anecdotally, we noticed that the fibroblast strains that responded poorly to exogenous administration of rhPPCA had high endogenous levels of cathepsin A (Figure 2D). This could represent a limiting factor for assessing small variations in NEU1 residual activity, particularly considering the restrictions posed by an in vitro ERT. Future studies on a larger number of patients with sialidosis type I may shed more light on how NEU1 and PPCA regulate each other and influence their reciprocal activities.

### 3.3. Romidepsin, a Class I HDAC Inhibitor, Increases NEU1 Levels

We have recently demonstrated that *NEU1* mRNA expression responds to histone deacetylase (HDAC) inhibitors and that *NEU1* levels are regulated by HDAC2, one of the 11 members of the HDAC family belonging to the classical class I HDAC [14]. Based on these findings, we wanted to test the effect of another Food and Drug Administration (FDA)-approved HDAC inhibitor, romidepsin, that targets more specifically class I HDAC [20,21,22], on all 12 type I sialidosis fibroblasts included in this study. Administration of this compound resulted in a striking and significant increased expression of the different mutant *NEU1* mRNAs (Figure 3A). This was accompanied by an equally significant increase of the corresponding residual enzyme activities and protein levels of the NEU1 mutant enzymes in all sialidosis type I fibroblasts (Figure 3B,C and Appendix A). Thus, romidepsin or any other HDAC inhibitors, may represent a potential alternative therapy for patients with sialidosis type I.

### 3.4. Betaine Increases NEU1 Levels in Sialidosis Fibroblasts.

An intriguing observation that caught our attention was made recently by the group of Ritva Tikkanen that demonstrated a slight increase in the residual activity of mutant aspartylglucosaminidase upon administration of betaine to fibroblasts from patients with AGA [15]. These results prompted us to investigate whether betaine might have a similar effect on the residual activity of NEU1 protein variants, and, in turn, hold therapeutic potential for type I sialidosis. Betaine treatment increased *NEU1* mRNA levels in several of the sialidosis fibroblasts (Figure 4A). This was accompanied by a modest but significant increase of the residual activity of NEU1 mutant proteins (Figure 4B), and by a measurable increase of protein levels, as determined by Western blot analysis (Figure 4C and Appendix A).

Although the biological mechanism(s) of action of betaine are not fully elucidated, its three methyl groups serve as methyl donors in multiple metabolic pathways [23]. In line with its involvement in methylation reactions, betaine has been reported to increase the levels of trimethylation of histone H3 on lysine 4 (H3K4me3), which is known to be associated with transcriptional activation of gene expression [24,25]. We, therefore, tested the levels of this histone mark in the betaine-treated sialidosis fibroblasts and found a marked increase of H3K4me3 in the majority of the patients‘ cells (Figure 4D and Appendix A), suggesting a potential influence of this compound in transcription.

### 3.5. Betaine Regimen Increases Neu1 Levels In Vivo

As proof of principle, we next tested the effects of betaine in vivo in two mouse models, which retain NEU1 residual activity. One was the above-mentioned *NEU1^–/–^; Neu1^V54M^* mouse [11], the other, (*Neu1^del654-659^*), is a new mouse model generated by targeting *NEU1* with the CRISPR/CAS9 technology. The latter model retains ~2% residual Neu1 activity and measurable levels of Neu1 mutant protein (~27%) in visceral organs, especially in the kidney, which normally expresses the highest levels of NEU1 in humans and mice (Appendix A). Both models represent valuable in vivo systems to test compounds aimed to increase Neu1 residual activity.

Mice of different ages (one month to one year) were treated with betaine for a month *ad libitum* and then sacrificed. To monitor the outcome, we assayed changes in Neu1 activity specifically in peripheral blood mononuclear cells (PBMCs) and measured total urinary sialic acids, because these standard assays in samples that can be obtained non-invasively mimic the way NEU1 activity would be monitored in sialidosis patients pre- and post-treatment. Betaine administration resulted in a measurable increase in Neu1 activity in the PBMCs (Figure 5A,B) in both mouse models and reduced the total amount of urinary sialic acids in the *Neu1^del654-659^* treated animals (Figure 5C). Reduction of sialyl-oligosacchariduria in the *NEU1^–/–^; Neu1^V54M^* mice could not be tested since these mice manifest this phenotype at nearly two years of age [11]. Remarkably, and paralleling the results obtained in the urine, we observed a tangible increase in Neu1 activity in the kidney of both the betaine treated mice (Figure 5D,E).

These encouraging results suggest that betaine represents a potential novel drug candidate for the treatment of sialidosis.

## 4. Discussion

Sialidosis type I is the normosomatic form of sialidosis with onset of clinical signs on average in the second decade of life and often restricted to myoclonus, ataxia and visual impairment. There is no curative therapy currently available for this disease and the number of diagnosed patients is growing steadily every year, despite the difficulty to reach differential diagnosis of patients with symptoms often mistaken for other neurological conditions. The medicinal drugs commonly prescribed to sialidosis patients can marginally ameliorate some of the disease symptoms and are accompanied by a variety of adverse effects. In the last years, there has been a renewed interest in gene therapy approaches, due to the development of safer and effective therapeutic vectors [26,27]. The latter would be the preferred approach for sialidosis type I patients, providing a stable long-lasting therapeutic correction. Unfortunately, due to what is still considered a small group of eligible patients, these therapeutic approaches may not be available for sialidosis patients in the clinic anytime soon. Therefore, it is imperative to find additional means to halt this disease and improve the quality of life of patients.

In this study we have tested conventional and unconventional therapeutic approaches for sialidosis type I and have identified several compounds that could potentially be beneficial to some of these patients. Until now, no studies have evaluated the effects of dietary supplements in ameliorating disease progression in sialidosis.

Exploiting the chaperone role of PPCA on NEU1, we first tested the effect of rhPPCA on the residual activity of several mutant forms of NEU1 in sialidosis fibroblasts. We found that exogenously administered rhPPCA was able to increase mutant enzyme activity in a limited number of patients’ cells. This may depend on the type of NEU1 mutations present in the different fibroblast strains, which likely impact the folding and lysosomal localization of the mutant proteins. In addition, we also noticed that endogenous PPCA levels in many of these patients’ cells are higher than control cells, likely limiting the effect of the exogenously added PPCA. These results raise the intriguing possibility that NEU1 and PPCA regulate each other’s activities.

In search of an alternative method to boost NEU1 activity, we opted for romidepsin, a highly potent, brain-permeable HDAC class I-specific inhibitor [20,22]. This choice was based on the striking increase of normal *NEU1* expression levels upon treatment of cells with HDAC inhibitors [14]. The response to romidepsin treatment was remarkably effective in all 12 sialidosis fibroblasts. Interestingly, some of the patients’ fibroblasts responded even better than control cells to this compound. These results suggest that some of the patients may have an abnormally low levels of histone acetylation at the *NEU1* locus or abnormal levels of HDAC. These findings merit future investigations, especially if such compounds are considered for therapy.

Until now romidepsin has been approved by the FDA only for the treatment of cancer, which could limit its use for the treatment of other metabolic diseases. However, it is worth mentioning that such inhibitors are natural constituents of commonly used dietary and herbal products [28,29,30]. These include cruciferous vegetables (broccoli and broccoli sprouts), rich in sulforaphane and other isothiocyanates, garlic, which contains diallyl disulfide, dietary fibers that release butyrate by fermentation in the colon, and lastly black cumin, monarda and thyme, which are enriched in thymoquinone. Complementing a normal diet with a balanced intake of these foods and food supplements could increase residual NEU1 activity and ameliorate specific phenotypes of sialidosis.

Betaine or trimethylglycine is a natural modified amino acid produced by the human body from choline oxidation, or readily assimilated by the consumption of dietary products that are enriched in this compound, such as grains, beets, spinach and shellfish [31,32,33,34]. Betaine is known to function as a methyl donor for the conversion of homocysteine to methionine via the enzymatic activity of the betaine-homocysteine-methyltransferase, primarily in the liver and kidney. In this capacity, betaine has been used in the clinic for the treatment of homocystinuria [23]. As an osmolyte, betaine increases the water retention of cells, replaces inorganic salts, and protects against osmotic shocks [35]. Betaine has been used with good results in a variety of experimental disease models, including AGA, alcohol liver disease, nonalcoholic fatty liver disease, obesity, diabetes and multiple sclerosis [15,24,25,36,37,38,39,40,41,42,43,44]. Although its mechanism of action is not fully understood, it is suggested to act primarily as methyl donor in several metabolic reactions, or to influence epigenetic marks. Our results show that betaine can enhance NEU1 activity in vitro and in vivo and also increase H3K4me3 levels, and point to a role of betaine in modulating the epigenetic landscape and, in turn, transcription. This epigenetic action of betaine should in principle translate into a higher expression of the protein. However, the location and the type of mutation(s) in the *NEU1* gene may still influence the enzymatic activity due to inefficient binding of the mutant protein to PPCA. This is supported by the differential response to betaine treatment that we have obtained in the different patients’ cells. Lastly, we cannot exclude, that betaine has also an effect on the stability of the NEU1 protein. Overall these results are in agreement with data obtained in piglets exposed to maternal supplementation of betaine [24,25], and in models of multiple sclerosis [45,46]. As expected for other LSDs, the residual NEU1 activity in sialidosis type I patients ranges from 1% to 5% of normal enzyme levels [47]. Therefore, even relatively small increases of NEU1 residual activity may have a positive impact on disease severity and symptomatology. Following the example of AGA in Finland, it is plausible that these natural compounds could be tested in future clinical trials for sialidosis type I.

Considering the spectrum of disease severity of patients with sialidosis type I, and the outcome of these studies, a careful assessment of the dietary habits of individual patients should be taken into account. It is predictable that patients who follow diet regimens rich in HDAC inhibitors and betaine may see an improvement in their quality of life.

In conclusion, because it is well established that diet plays a major role in the prevention and control of diseases like cancer and neurodegenerative disorders, we believe that the same type of proactive measures should be applied to sialidosis type I and, in general, to all attenuated forms of LSDs.

## Figures and Tables

**Figure 1 jcm-09-00695-f001:**
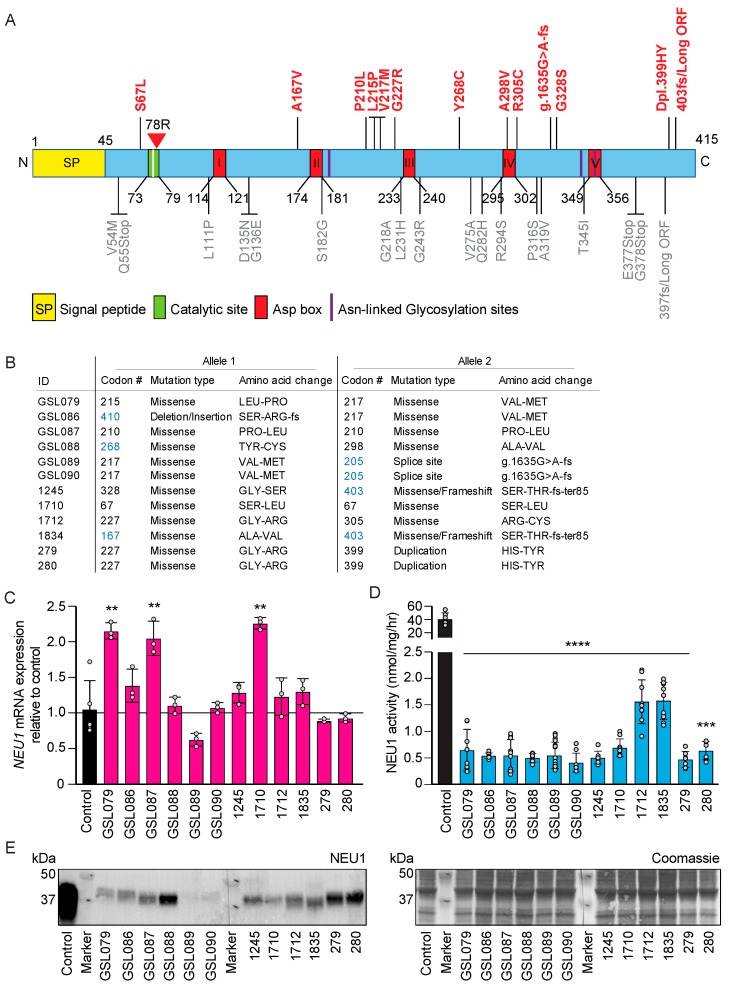
Molecular and Biochemical Characteristics of Sialidosis Patients (**A**) Schematic representation of neuraminidase 1 (NEU1) mutations within the primary structure of NEU1. Mutations indicated in red are those belonging to the sialidosis type I patients used in this study. Conserved Asp-box motifs are numbered I–V, and amino acid positions are indicated. The signal peptide (SP) is marked by a yellow box. (**B**) Biallelic NEU1 mutations in the cohort of sialidosis type I patients used in these studies. Mutations in blue were not previously described. (**C**) Levels of NEU1 mRNA in sialidosis fibroblasts were calculated relative to those in control cells (*n* = 3). (**D**) NEU1 residual activity assayed in sialidosis type I fibroblasts (*n* ≥ 6). (**E**) Western blot analysis of NEU1 mutant proteins in lysates of sialidosis type I cells; short and long exposure of a representative immunoblot probed with anti-human NEU1 antibody. Coomassie stained immunoblot used as loading control (*n* = 3). Graphs are presented as mean ± SD. Statistical analysis was performed using Student *t*-test. ** *p* < 0.01, *** *p* < 0.001, **** *p* < 0.0001.

**Figure 2 jcm-09-00695-f002:**
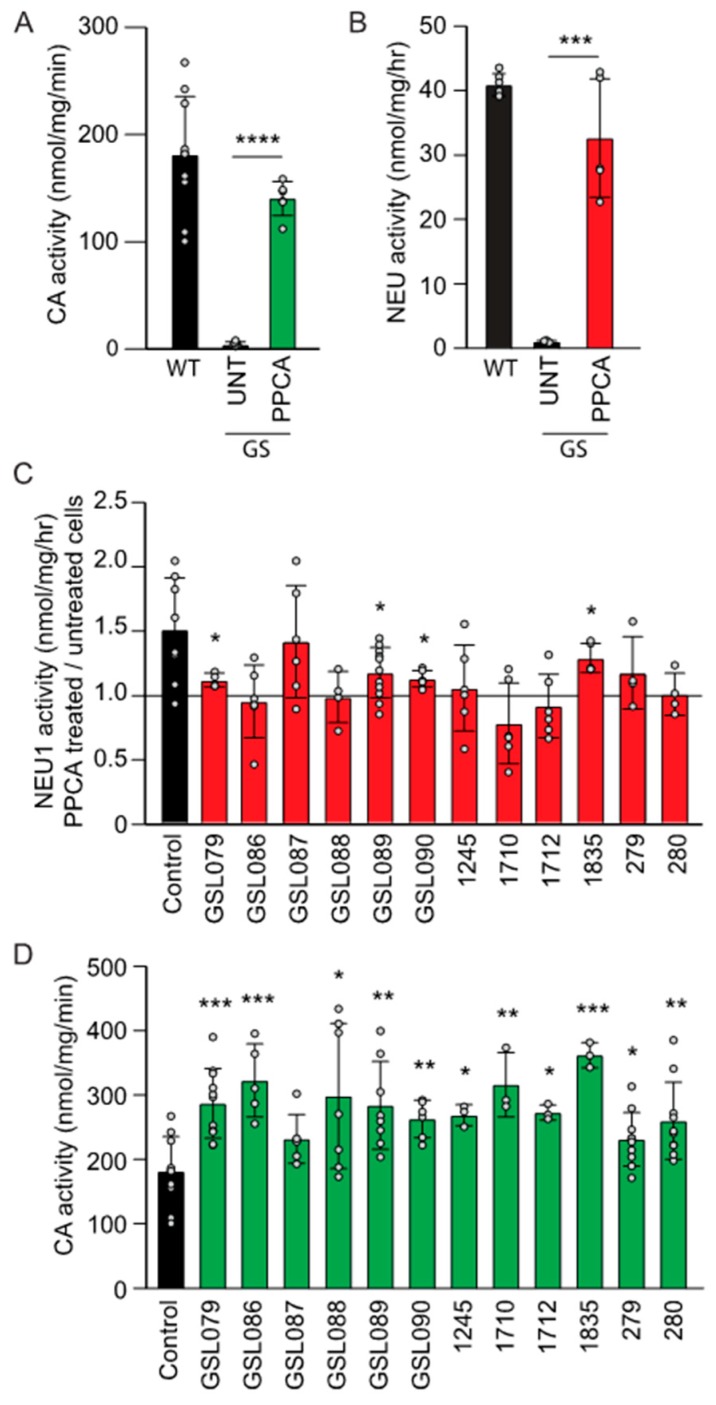
Sialidosis type I fibroblasts respond to treatment with human recombinant protein/cathepsin A (PPCA). (**A**) cathepsin A and (**B**) NEU1 activities assayed in galactosialidosis fibroblasts after addition of recombinant human protective protein/cathepsin A (rhPPCA) (*n* ≥ 5) (**C**) Sialidosis type I fibroblasts (*n* ≥ 4) assayed for NEU1 activity after addition of rhPPCCA. (**D**) Cathepsin A activity measured in sialidosis type I fibroblasts (*n* ≥ 3). Graphs are presented as mean ± SD. Statistical analysis was performed using Student *t*-test. * *p* < 0.05, ** *p* < 0.01, *** *p* < 0.001, **** *p* < 0.0001.

**Figure 3 jcm-09-00695-f003:**
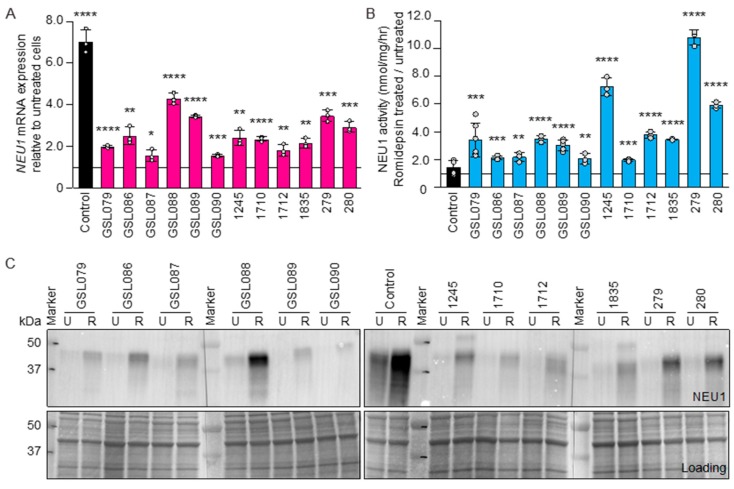
Romidepsin increases residual NEU1 activity in sialidosis type I fibroblasts. (**A**) Levels of NEU1 mRNA in sialidosis fibroblasts treated with romidepsin. Values were calculated relative to those in control cells (*n* = 3). (**B**) NEU1 activity assayed in sialidosis type I fibroblasts (*n* = 3) after treatment with romidepsin. (**C**) Representative immunoblot of romidepsin-untreated (U) and -treated (R) sialidosis fibroblasts probed with anti-human NEU1 antibody. Coomassie-stained immunoblot used as the loading control. Graphs are presented as mean ± SD. Statistical analysis was performed using Student *t*-test. * *p* < 0.05, ** *p* < 0.01, *** *p* < 0.001, **** *p* < 0.0001.

**Figure 4 jcm-09-00695-f004:**
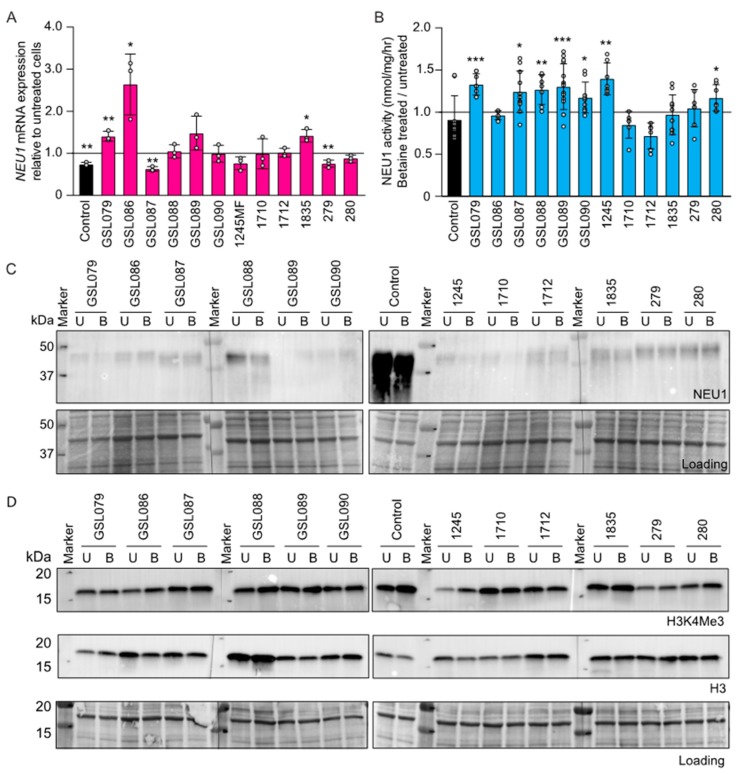
Betaine modulates NEU1 levels in sialidosis type I fibroblasts. (**A**) Levels of NEU1 mRNA in sialidosis fibroblasts treated with betaine. Values were calculated relative to those in control cells (*n* = 3). (**B**) NEU1 activity assayed in sialidosis type I fibroblasts (*n* ≥ 5) after treatment with betaine. (**C**) Representative immunoblot of betaine-untreated (U) and -treated (**B**) sialidosis fibroblasts probed with human anti-NEU1 antibody. Coomassie-stained immunoblot used as the loading control (*n* = 3). (**D**) Representative immunoblot of betaine-treated sialidosis type I fibroblasts probed with anti-Histone H3 lysine 4 trimethylation (H3K4me3) and anti-histone H3 (H3) antibodies. Coomassie-stained immunoblot used as the loading control (*n* = 3). Graphs are presented as mean ± SD. Statistical analysis was performed using Student *t*-test. * *p* < 0.05, ** *p* < 0.01, *** *p* < 0.001.

**Figure 5 jcm-09-00695-f005:**
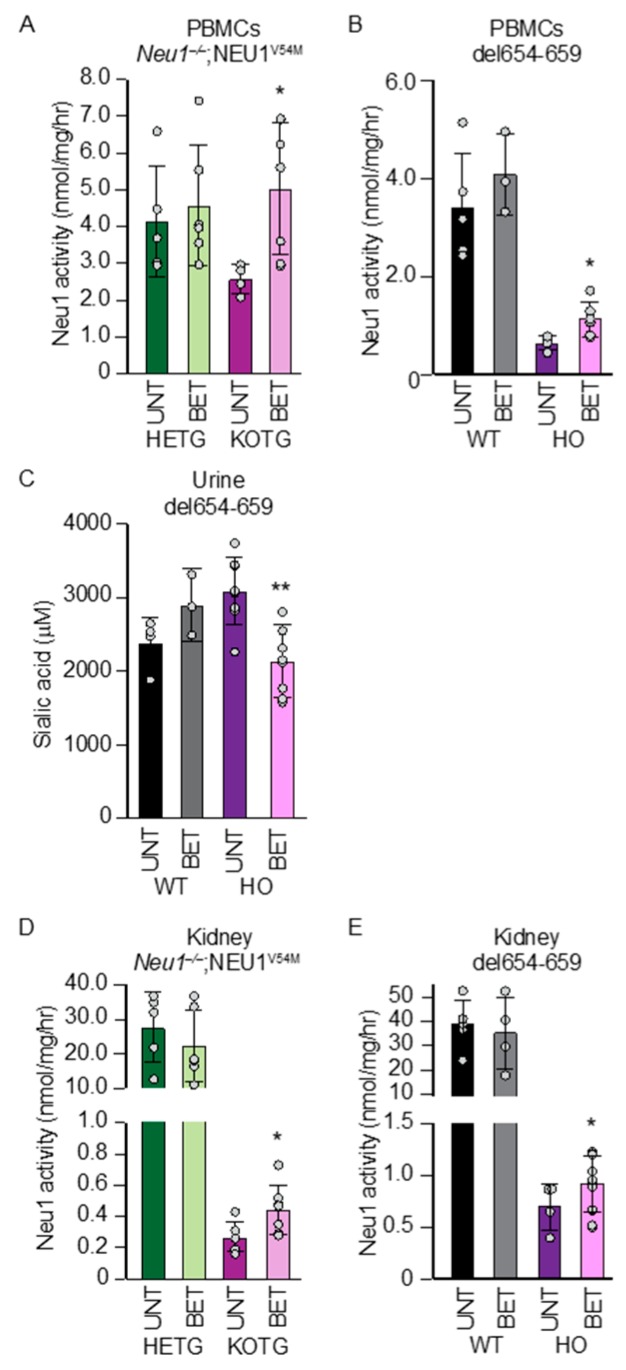
Betaine modulates Neu1 levels in vivo in mouse models with Neu1 residual activity. (**A**,**B**) Neu1 activity assayed in peripheral blood mononuclear cells (PBMCs) isolated from *Neu1^+/−^; NEU1^V54M^* (HETG), *Neu1^−/−^; NEU1^V54M^* (KOTG) (**A**, *n* ≥ 4) and mice homozygous for the del654-659 (HO) (**B**, *n* ≥ 3). (**C**, *n* ≥ 3) sialic acid content in the urine of untreated and betaine-treated del654-659 (HO). Neu1 activity assayed in the kidneys of untreated and betaine-treated KOTG (**D**, *n* ≥ 5) and del654-659 (HO) (**E**, *n* ≥ 4) respectively. Graphs are presented as mean ± SD. Statistical analysis was performed using Student *t*-test. * *p* < 0.05, ** *p* < 0.01.

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
