# Peer review of "Conventional and Unconventional Therapeutic Strategies for Sialidosis Type I"

_jcm, 2020, doi:10.3390/jcm9030695_

Round 1
Reviewer 1 Report
The manuscript by Mosca et al. describes the effects of recombinant human PPCA, romidepsin, and betamine on NEU1 mRNA and protein levels and catalytic activity in fibroblasts from patients with sialidosis type I, and mice genetically engineered in the Neu1 locus to mimic the human disease. In general, the manuscript is well-written and the conclusions are supported by the results presented. The following revisions are suggested before final acceptance or publication in the Journal of Clinical Medicine.
Major
Fig. 4. It is unclear why the effect of betamine on NEU1 mRNA levels does not correlate with NEU1 activity or protein levels. For example, fibroblasts from patient GSL086 have the greatest increase in NEU1 mRNA under the betamine stimulus (Fig 4A), but no increases in NEU1 activity (Fig 4B) or protein (Fig 4C). Could the authors please comment on this apparent discrepancy? The effect of increasing NEU1 mRNA, protein, and catalytic activity appears more pronounced and consistent using romidepsin treatment compared with betamine. Can the authors please comment why they chose to assess the effects of betamine rather than romidepsin in vivo?
Minor
Line 194, typo: @l Line 229: add Bonten, Arts 2000 to references
Author Response
Comments and Suggestions for Authors
The manuscript by Mosca et al. describes the effects of recombinant human PPCA, romidepsin, and betamine on NEU1 mRNA and protein levels and catalytic activity in fibroblasts from patients with sialidosis type I, and mice genetically engineered in the Neu1 locus to mimic the human disease. In general, the manuscript is well-written and the conclusions are supported by the results presented. The following revisions are suggested before final acceptance or publication in the Journal of Clinical Medicine.
R: We thank this reviewer for his positive comments and his/her support.
Major
Fig. 4. It is unclear why the effect of betamine on NEU1 mRNA levels does not correlate with NEU1 activity or protein levels. For example, fibroblasts from patient GSL086 have the greatest increase in NEU1 mRNA under the betamine stimulus (Fig 4A), but no increases in NEU1 activity (Fig 4B) or protein (Fig 4C). Could the authors please comment on this apparent discrepancy?
R: The effect of betaine on increasing slightly NEU1 mRNA is likely epigenetic, and in principle should translate into a higher expression of the protein. However, the location and type of mutation may still affect the enzymatic activity due also to inefficient binding to PPCA. This is substantiated by the variable response to betaine treatment we have obtained in the different patients’ cells.
The effect of increasing NEU1 mRNA, protein, and catalytic activity appears more pronounced and consistent using romidepsin treatment compared with betamine. Can the authors please comment why they chose to assess the effects of betamine rather than romidepsin in vivo?
R: We agree with the reviewer that romidepsin would be a great option for patients with sialidosis. The effect of romidepsin on NEU1 is very robust and this translates into a sustained increase in NEU1 enzyme activity. Unfortunately, romidepsin is only approved (because of toxicity and multiple side effects) and used for the treatment of certain type of cancers. We choose for betaine because is a natural compound, readily available over the counter or present in commonly used foods.
Minor
Line 194, typo: @l Line 229: add Bonten, Arts 2000 to references
R: the reference was added to the list.
Reviewer 2 Report
This paper looks at alternative therapeutic options for sialidosis type 1.
- Early in the paper the authors discuss ERT, gene therapy and pharmacologic chaperone therapy with PPCA - the authors should discuss these in more detail to ensure the paper is reflective of current understanding and therapy development for sialidosis.
There is no discussion on level of efficacy proposed by the alternative therapeutics and also next steps after fibroblast / mouse models. What new / follow-up experiments are planned? Fig 5 suggests all other interventions other than betaine lacked statistical significance - is this accurate? It also appears only del654-659 had a significant effect, can the authors explain why Neu1-/- did not have a significant effect? In addition, the bars show significant overlap in error / confidence, can the authors outline in detail their statistical methods?
Author Response
Comments and Suggestions for Authors
This paper looks at alternative therapeutic options for sialidosis type 1.
- Early in the paper the authors discuss ERT, gene therapy and pharmacologic chaperone therapy with PPCA - the authors should discuss these in more detail to ensure the paper is reflective of current understanding and therapy development for sialidosis.
There is no discussion on level of efficacy proposed by the alternative therapeutics and also next steps after fibroblast / mouse models. What new / follow-up experiments are planned?
R: We have now added these points in the discussion section.
Fig 5 suggests all other interventions other than betaine lacked statistical significance - is this accurate?
R: We have added statistical significance to all our graphs.
It also appears only del654-659 had a significant effect, can the authors explain why Neu1-/- did not have a significant effect?
R: We have now performed additional experiments and increased the number of animals tested and calculated statistical significance.
In addition, the bars show significant overlap in error / confidence, can the authors outline in detail their statistical methods?
R: The detailed statistical analyses are described in the Materials and Methods section of the manuscript. We have analyzed our data using the Student t-test two-tailed unpaired.
Reviewer 3 Report
This manuscript reports on potential therapeutic interventions for patients with Sialidosis Type I. Globally, the manuscript is very good. Not only was the study well designed, but the manuscript was written in a very sharp and concise manner, which allows for the reader to easily follow the procedures and evaluate the resulting conclusions.
It is important that these data are published as new patients who are diagnosed with sialidosis type I may be able to improve their quality of life with these manageable interventions, although more work will need to be done to determine their full effect on patients.
There are a few minor revisions that need to be made to the manuscript:
Line 42: Change protectice to protective
Line 49: Change diseases to disease
Line 61: Remove words at least, provide age range if available
Line 62: add the word 'do' in front of 'not always correlate...'
Line 66: Change undistinguishable to indistinguishable
Provide reference for statement in lines 65-68 regarding misdiagnosis.
Lines 69-71: Please add additional information regarding variants identified (ex: pathogenic, compound heterozygous variants).
Lines 73-75: Please state estimated incidence of sialidosis; can you use data from gnomeAD browser and make calculation with Hardy Weinberg equation to calculate what the actual incidence may be since you are suggesting the incidence in the general population is much higher?
Line 79: Is the amino acid substitution is homozygous in the mouse model?
Line 91: Add reference
Line 99: Suggest to change AGA patients to patients with AGA.
Line 186: Recommend to change ddH2O to ddH2O
Lines 215-216: Please use HGNC guidelines for nomenclature of variants listed in the text.
Line 229: The reference provided is in a different format than the other references (Name and year; not numerical)
Line 285: Please change age to ages; also, if you can the range of ages with min and max that would be helpful.
Author Response
Comments and Suggestions for Authors
This manuscript reports on potential therapeutic interventions for patients with Sialidosis Type I. Globally, the manuscript is very good. Not only was the study well designed, but the manuscript was written in a very sharp and concise manner, which allows for the reader to easily follow the procedures and evaluate the resulting conclusions.
It is important that these data are published as new patients who are diagnosed with sialidosis type I may be able to improve their quality of life with these manageable interventions, although more work will need to be done to determine their full effect on patients.
R: We thank this reviewer for his/her positive comments and support of this work
There are a few minor revisions that need to be made to the manuscript:
Line 42: Change protectice to protective
R: We have corrected this misspelling.
Line 49: Change diseases to disease
R: We have made this correction.
Line 61: Remove words at least, provide age range if available
R: We have made the suggested correction. The age range is difficult to estimate because many of the patients are misdiagnosed at the onset of the symptoms.
Line 62: add the word 'do' in front of 'not always correlate...'
R: We have made this correction.
Line 66: Change undistinguishable to indistinguishable
R: We have corrected this misspelling.
Provide reference for statement in lines 65-68 regarding misdiagnosis.
R: We have added a reference for this statement.
Lines 69-71: Please add additional information regarding variants identified (ex: pathogenic, compound heterozygous variants).
R: We have added this information to the text.
Lines 73-75: Please state estimated incidence of sialidosis; can you use data from gnomeAD browser and make calculation with Hardy Weinberg equation to calculate what the actual incidence may be since you are suggesting the incidence in the general population is much higher?
R: Unfortunately, it is impossible to separate the incidence of type I from type II sialidosis. The former patients have likely a higher incidence in the population than type II patients.
Using the Hardy Weinberg equation, the calculated (estimated) frequency for sialidosis (combining type I and type II alleles) is 1:33,339,840 in the general population even rarer than what reported in the literature. We have now added in the text the incidence for the disease reported by Caciotti et al (1:250,000 to 1:2,000,000 live births)
Line 79: Is the amino acid substitution is homozygous in the mouse model?
R: This amino acid substitution is present in homozygosity in the mouse model
Line 91: Add reference
R: We have added a reference to line 91.
Line 99: Suggest to change AGA patients to patients with AGA.
R: We have changed the text accordingly.
Line 186: Recommend to change ddH2O to ddH2O
R: We have changed ddH2O to ddH2O.
Lines 215-216: Please use HGNC guidelines for nomenclature of variants listed in the text.
R: All the variants now are referred to by using the HGNC guidelines.
Line 229: The reference provided is in a different format than the other references (Name and year; not numerical)
R: We have changed the format of the reference in L229.
Line 285: Please change age to ages; also, if you can the range of ages with min and max that would be helpful.
R: We have made the change and added the age range (1 month to 1 year) to the text.
